**Subject Area:**
microbiology/immunology/cellular biology

autophagy, arboviruses, flaviviruses, alphaviruses, dengue virus, West Nile virus

**Authors for correspondence:**
Jolanda M. Smit
e-mail: jolanda.smit@umcg.nl
Fulvio Reggiori
e-mail: f.m.reggiori@umcg.nl

# Role of autophagy during the replication and pathogenesis of common mosquito-borne flavi- and alphaviruses

Liliana Echavarria-Consuegra[1], Jolanda M. Smit[1] and Fulvio Reggiori[2]

[1]Department of Medical Microbiology and Infection Prevention, and [2]Department of Cell Biology, University of Groningen, University Medical Center Groningen, Groningen, The Netherlands

(iD) FR, 0000-0003-2652-2686

Arboviruses that are transmitted to humans by mosquitoes represent one of the most important causes of febrile illness worldwide. In recent decades, we have witnessed a dramatic re-emergence of several mosquito-borne arboviruses, including dengue virus (DENV), West Nile virus (WNV), chikungunya virus (CHIKV) and Zika virus (ZIKV). DENV is currently the most common mosquito-borne arbovirus, with an estimated 390 million infections worldwide annually. Despite a global effort, no specific therapeutic strategies are available to combat the diseases caused by these viruses. Multiple cellular pathways modulate the outcome of infection by either promoting or hampering viral replication and/or pathogenesis, and autophagy appears to be one of them. Autophagy is a degradative pathway generally induced to counteract viral infection. Viruses, however, have evolved strategies to subvert this pathway and to hijack autophagy components for their own benefit. In this review, we will focus on the role of autophagy in mosquito-borne arboviruses with emphasis on DENV, CHIKV, WNV and ZIKV, due to their epidemiological importance and high disease burden.

## 1. Introduction

### 1.1. The epidemiology of arboviruses

Arbovirus (arthropod-borne virus) is an ecological term that groups viruses transmitted to their hosts through the bite of blood-feeding arthropods, such as ticks, mosquitoes and sandflies [1]. It comprises over 500 viruses, which are classified into six main taxonomic groups: family *Togaviridae* (genus *Alphavirus*), family *Flaviviridae* (genus *Flavivirus*), order Bunyavirales (families *Orthobunyavirus*, *Nairovirus* and *Phlebovirus*), family *Rhabdoviridae* (7 genera), family *Orthomyxoviridae* (genus *Thogotovirus*) and family *Reoviridae* [2,3]. Some of these viruses have become major human pathogens, due to their rapid dispersal around the world or their persistence throughout the years. This is primarily linked to the expansion of the habitats of their vectors as a consequence of global warming, unplanned urbanization and unintentional transport [4]. In recent decades, we have witnessed a dramatic re-emergence of arboviruses transmitted to humans by mosquitoes of the *Culex* spp. and/or *Aedes* spp., such as dengue virus (DENV), West Nile virus (WNV), chikungunya virus (CHIKV) and Zika virus (ZIKV), which are currently spread in both the western and eastern hemispheres [5]. It has been estimated that the population at risk of DENV and CHIKV infection is approximately 2.5 and 1.3 billion people, respectively [6–8].

 Most individuals infected with mosquito-borne arboviruses remain asymptomatic. During symptomatic infections, however, individuals often develop an undifferentiated febrile illness, accompanied by (severe) headache, body aches, joint pains, vomiting, diarrhoea and rash [9]. In the case of DENV, for example, an estimated 390 million individuals are infected each year and approximately

royalsocietypublishing.org/journal/rsob    Open Biol. 9: 190009

50–100 million individuals develop a symptomatic infection [10]. CHIKV infection, on the other hand, is associated with a relatively high symptomatic attack rate, as 50–97% of the infected individuals develop a clinically apparent disease [11]. Additionally, more severe clinical manifestations have been reported in a small subset of infected people, such as meningitis or encephalitis (e.g. WNV), debilitating chronic arthralgia (e.g. CHIKV), vascular leak and haemorrhage (e.g. DENV), or congenital malformations and microcephaly (e.g. ZIKV) [12,13]. In most situations, symptoms resolve without complications, yet prolonged fatigue, depression, chronic pain and permanent effects in the central nervous system (CNS) have been reported for some of these viruses [14,15]. In rare cases, arbovirus infections lead to death [14,15].

Despite the global threat of DENV, WNV, ZIKV and CHIKV, vaccines and treatment possibilities for the infections caused by these viruses are scarce. Treatments remain palliative as no specific antivirals are available thus far [16–18]. A substantial number of studies have, however, explored several treatment strategies, but currently, none of them is approved for human use [19]. Effective prophylactic immunization exists for few arboviruses such as Japanese encephalitis virus and yellow fever virus [20]. In addition, multiple efforts have been made regarding the development of DENV, ZIKV, WNV and CHIKV vaccines. Dengvaxia (also known as CYD-TDV) developed by Sanofi Pasteur has recently become the first approved DENV vaccine [21,22]. Although it has been licensed in several countries in South and Central America, and in the Philippines, the introduction of this vaccine to mass immunization programmes is currently not recommended by the World Health Organization due to safety issues [23]. In the case of CHIKV, several vaccine candidates have been developed, including a recombinant measles virus expressing CHIKV antigens and a virus-like particle vaccine, which have successfully completed phase I clinical trials [24,25]. Given the high disease burden in particular of DENV and CHIKV, it is of utmost importance to further develop promising existing strategies and to explore new therapeutic and immunization methodologies to combat these viruses. Understanding the arbovirus virus–host interaction is crucial for this goal.

## 1.2. Replication cycle of flavi- and alphaviruses

DENV, WNV and ZIKV are enveloped single-stranded positive-sense RNA (ssRNA+) viruses that belong to the *Flavivirus* genus. The genomic RNA is packaged by capsid (C) proteins to form the nucleocapsid [26]. The flaviviral genome is 10–12 kb long and it encodes for a single open reading frame (ORF) [27]. The flavivirus ssRNA+ has a 5′-cap structure but lacks a 3′-poly(A) tail [27]. It also contains 5′- and 3′-untranslated regions (UTR) that fold into secondary structures and are conserved among divergent flaviviruses [27–31]. The nucleocapsid is surrounded by a host cell-derived envelope in which two transmembrane proteins, the membrane (M) protein and the envelope (E) protein, are inserted [32,33].

During infection, the E protein mediates the attachment of virus particles to the cell surface (figure 1). Multiple receptors have been identified and their usage depends on the cell type and virus [34]. Virus recognition is followed by internalization of the virion through endocytosis and subsequent fusion between the membrane of the viral particle and the limiting membrane of late, Ras-related protein 7A (RAB7A)-positive

acidic endosomes, facilitated by the E protein [35–38] (figure 1). Once the RNA is delivered to the cytoplasm, the ssRNA+ is translated by ribosomes associated with the rough endoplasmic reticulum (ER) [39].

RNA translation generates a polyprotein of approximately 370 kDa in length that is inserted into the ER membrane and cleaved co- and post-translationally by viral and cellular proteases, into the individual proteins: the E, C and precursor M (prM) structural proteins, and the NS1, NS2A/B, NS3, NS4A/B, NS5 non-structural (NS) proteins. Extensive ER-derived membrane rearrangements are induced by the viral proteins NS4 and NS3, which serve as scaffolds for the assembly of viral replication complexes [40,41] (figure 1). The NS proteins are required for RNA replication and pathogenesis [27,42]. For instance, DENV NS2A, NS2B3, NS4A, NS4B and NS5, and WNV NS1 and NS4B are involved in immune evasion [43]. The viral NS5 polymerase synthesizes new ssRNA+ through the generation of an ssRNA− intermediary strand, and this can be used for new rounds of translation or as a substrate for encapsidation in progeny virions (figure 1). During encapsulation, viral RNA is packaged into the nucleocapsid by interaction and assembly of multiple copies of the C protein [44]. The envelope prM and E proteins form heterodimers that are oriented into the lumen of the ER and associate into trimers to create a curved surface lattice, which guides the budding of the nucleocapsid into the ER to form immature viral particles [45] (figure 1). These immature particles are transported through the secretory pathway to the trans-Golgi network (TGN), where the prM/E envelope proteins undergo conformational changes thereby allowing the host protease furin to process prM into M, which drives maturation of the virus [33,46]. Progeny flavivirus particles are finally secreted from the cells by exocytosis [13] (figure 1).

CHIKV, a member of the *Alphavirus* genus, has a ssRNA+ genome of 11.8 kb. The RNA is packaged by the capsid protein (C) to form a nucleocapsid. The nucleocapsid is surrounded by an envelope wherein the two transmembrane glycoproteins, E1 and E2, are anchored [47]. The CHIKV genome resembles eukaryotic mRNAs as it has a 5′-cap structure and a 3′-poly-adenine tail [48]. It also has 5′- and 3′-non-translatable regions (NTR) composed of 76 nucleotides and 526 nucleotides, respectively [48]. Unlike flaviviruses, the CHIKV genome contains two ORFs, separated by a 68-nucleotide long untranslated junction region [48].

The E2 protein mediates binding of the virus to cell surface receptors, which is followed by internalization of the virus via clathrin-mediated endocytosis and subsequent E1-mediated fusion between the virion membrane and the limiting membrane of acidic early, RAB5A-positive endosomes [49] (figure 2). The subsequent disassembly of the capsid is thought to occur upon binding of the C protein to the large ribosomal subunit, which leads to the release of the viral RNA [50] (figure 2).

Upon release of the ssRNA+ into the cytoplasm, the 5′-ORF is rapidly translated into a polyprotein (P1234), the viral replicase, which is composed by the nsP1, nsP2, nsP3 and nsP4 NS proteins of the virus [51] (figure 2). First, P1234 is cleaved in *cis* by nsP2 to generate P123 and nsP4 [52], which leads to the formation of an unstable replication complex that synthesizes ssRNA− intermediates in structures near the plasma membrane known as spherules [53]. Later in infection, these spherules are relocated to the limiting membrane of small cytoplasmic vesicles, giving rise to

royalsocietypublishing.org/journal/rsob   Open Biol. **9**: 190009

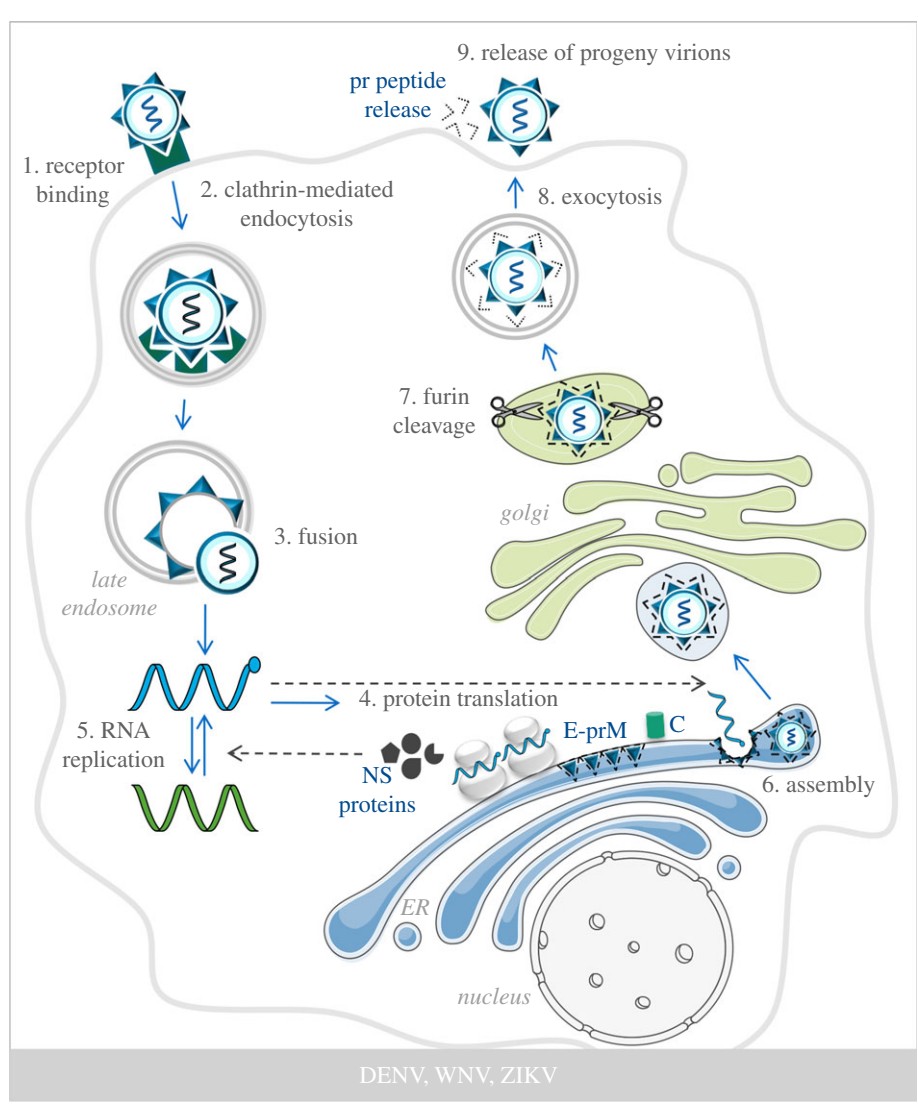

**Figure 1.** Flavivirus replication cycle. Flavivirus infection starts with the binding of the virion to cell receptors (step 1), which subsequently triggers the internalization of the viral particle via clathrin-mediated endocytosis (step 2). The acidic environment of late endosomes triggers the fusion of the virion with the limiting membrane of this organelle, resulting in the release of the genomic RNA into the cell cytoplasm (step 3). Translation of the viral RNA generates a polyprotein that is proteolytically cleaved into the non-structural (NS) and the structural proteins (step 4). NS proteins facilitate RNA replication leading to the formation of ssRNA− (green) and ssRNA+ (blue) transcripts (step 5). Progeny ssRNA+ is packaged by the capsid protein (C) to form the nucleocapsid. Viral assembly takes place in the ER (step 6), resulting in immature virions that are transported to the TGN through the secretory pathway, where furin-mediated cleavage of prM into M generates mature viral particles (step 7) that are released extracellularly by exocytosis (steps 8 and 9). The pr peptides dissociate from the virions once those are in the extracellular milieu.

cytopathic viral replication vacuoles [54]. Once P123-nsP4 levels reach a stoichiometric threshold, the polyprotein is further cleaved to generate the individual NS proteins. Thereafter, the synthesis of ssRNA+ and subgenomic RNA (sgRNA) from the 3'-ORF [55,56] is initiated. Herein, the untranslated junction between the two ORFs participates as an internal transcription promotor of the sgRNA [48].

Translation of approximately 5 kb sgRNA generates a second polyprotein that produces the structural proteins [48] (figure 2). Once the C protein is translated, it auto-cleaves and a signal sequence in E3 directs the translocation of the remaining structural polyprotein (E3, E2, 6 K or TF and E1) into the ER membrane [57]. The C protein subsequently recognizes specific motifs in the 5'-end of the newly synthesized ssRNA+ to form nucleocapsid-like structures [58]. Meanwhile, host proteases catalyse the cleavage of the individual structural proteins to generate pE2 (fused E3–E2), 6 K or TF and E1 [59].

pE2 and E1 heterodimers undergo post-translational modifications and are transported through the TGN, where furin-mediated cleavage of pE2 into E2 and a soluble E3 peptide leads to the formation of E2–E1 heterodimers that are directed to the plasma membrane [60] (figure 2). Subsequent interaction of E2 proteins with a newly formed nucleocapsid drives virus assembly and budding from the plasma membrane [48] (figure 2). Although the function of 6 K protein in the replication cycle of alphaviruses is not fully understood, it is thought, among other functions, to interact with E1 and pE2 to regulate their trafficking to the plasma membrane [61]. The TF protein is generated from a ribosomal frameshift that occurs during translation of the 6 K gene and is believed to mediate CHIKV assembly and release, although its full function remains to be determined [62].

## 1.3. Autophagy

Autophagy is a catabolic pathway that is highly conserved among eukaryotes, in which cytoplasmic components, including organelles, long-lived proteins and protein

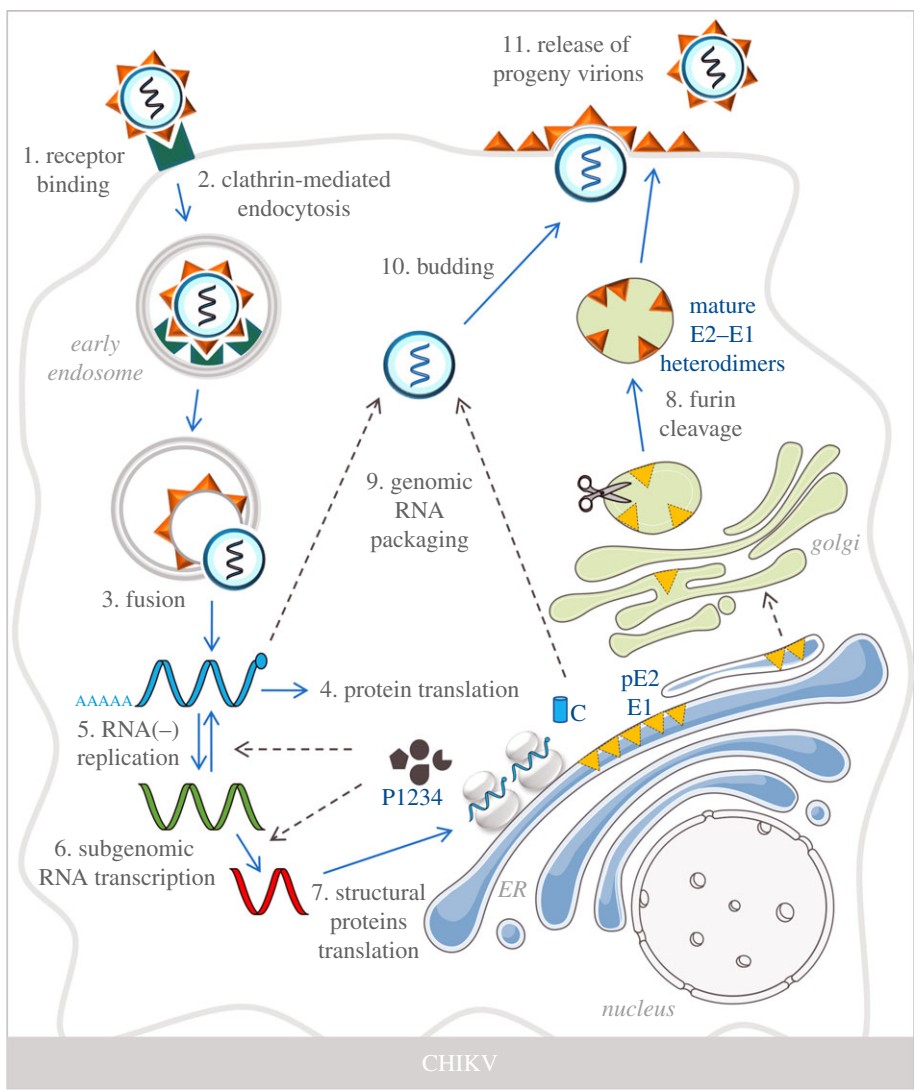

**Figure 2.** CHIKV replication cycle. CHIKV infection is initiated by the binding of the viral particle to cell receptors (step 1), which triggers the internalization of the virion via clathrin-mediated endocytosis (step 2). Subsequent fusion of the viral particle with the early endosome limiting membrane leads to the cytoplasmic release of the genomic RNA release (step 3). Genomic RNA is initially translated from the 5′-ORF into the viral replicase (P1234) (step 4), which replicate the ssRNA− (green) and the ssRNA− (blue) (step 5). The viral replicase also replicates the subgenomic RNA from the 3′-ORF (step 6), which serves as the template for the translation of structural proteins (step 7). The structural pE2 and E1 proteins are inserted into the ER and they are first processed in this organelle and then in the TGN, where furin-mediated proteolytical cleavage generates mature E2–E1 heterodimers that are exported to the plasma membrane (step 8). Genomic RNA is packaged by the C protein (step 9) and by interacting with the E2–E1 heterodimers, initiate the budding of the viral particle from the plasma membrane (step 10) to produce progeny virions (step 11).

complexes/aggregates are delivered into lysosomes for degradation and recycling of their basic components [63]. Three main types of autophagy are recognized in mammals: (1) macroautophagy, which involves the formation of double-membrane vesicles known as autophagosomes; (2) microautophagy, where the cytosolic material is directly engulfed through invagination of the lysosome limiting membrane; and (3) chaperone-mediated autophagy, in which proteins with a specific targeting motif are recognized by the cytosolic chaperone heat-shock cognate protein of 70 kDa (HSC70) and translocated into lysosomes through a channel formed on the surface of these organelles by LAMP2A [64,65]. Although these three pathways collectively support the overall intracellular autophagic activity, macroautophagy is the process that has been best characterized so far.

Macroautophagy, hereafter referred to as autophagy, contributes to the maintenance of cellular homeostasis by providing a mechanism for protein and organelle quality control. As a result, it plays a crucial role in numerous

physiological processes and pathological situations, such as cell development and cell differentiation, post-natal survival, immune response, neurodegenerative diseases, cancer, ageing and inflammation [66–68]. Autophagy is usually considered as a rather non-selective bulk degradation pathway, yet, it has become clear that it also contributes to intracellular homeostasis by selectively turning over specific substrates [69]. Distinctive terms have been coined to describe these types of selective autophagy, including mitophagy (mitochondria), lipophagy (lipid droplets), aggrephagy (aggregated proteins), pexophagy (peroxisomes), ribophagy (ribosomes), reticulophagy (ER) and xenophagy (pathogens) [70]. Autophagy is induced in response to a variety of cellular stressors, including nutrient deprivation and viral infections [71]. During starvation, the nutrient and energy-sensing kinases mechanistic target of rapamycin (mTOR) complex 1 (mTORC1) and the AMP-activated protein kinase (AMPK) directly regulate autophagy initiation (see below) [72]. Viral infection induces ER and oxidative stress, which in turn can

**Table 1.** Key proteins involved in autophagosome formation and its fusion with lysosomes in mammalian cells.

| step of autophagy | functional cluster | | components |
|---|---|---|---|
| initiation | ULK kinase complex | | ULK1-2 |
| | | | ATG13 |
| | | | FIP200 |
| | | | ATG101 |
| | autophagy-specific PI3 K-III complex | | VPS34 |
| | | | BECN1 |
| | | | p150 |
| | | | ATG14 L |
| | | | AMBRA1 |
| | ATG9A trafficking system | | WIPI1-4 |
| | | | ATG2A-B |
| | | | ATG9A |
| elongation | ubiquitin-like conjugation systems | ATG12 conjugation system | ATG7 |
| | | | ATG10 |
| | | | ATG16L1 |
| | | | ATG5 |
| | | | ATG12 |
| | | LC3 conjugation system | ATG4A-D |
| | | | LC3A-C/GAPARAP /GABARAPL1-L2 |
| | | | ATG7 |
| | | | ATG3 |
| fusion | CCZ1-MON1A | | |
| | RAB7 | | |
| | HOPS | | VPS11/VPS16/VPS18/VPS33A |
| | | | VPS39/VPS41 |
| | SNAREs | | STX17/VAMP8/ |
| | | | SNAP-29/YKT6 |
| cargo degradation | lysosomal enzymes | | cathepsin B, L, D and other hydrolases |

also trigger autophagy [73,74]. Upon ER stress, cells activate a series of adaptive mechanisms known as the unfolded protein response (UPR), to cope with the accumulation of misfolded proteins [75]. The UPR promotes the transcription of multiple groups of genes, including several of those involved in autophagy [76]. On the other hand, reactive oxygen species (ROS) production can directly activate autophagy (through mTORC1), to eliminate the source of oxidative stress and protect cells from oxidative damage [73]. Besides induction through cellular stress, autophagy can also be activated by the expression of several viral proteins [74,77].

The canonical form of autophagy is governed by five major functional clusters of proteins (table 1), which are composed by the so-called autophagy-related (ATG) proteins and work in concert in four sequential steps: (1) initiation and *de novo* formation of the phagophore (or isolation membrane), (2) elongation and closure of the phagophore to generate an autophagosome; (3) autophagosome–lysosome fusion and (4) cargo degradation and cytosolic recycling of the resulting metabolites (figure 3) [68,78]. Most of the ATG proteins that participate in these steps are localized in the cytoplasm and only associate with the forming autophagosomes upon autophagy induction [79]. This characteristic can be exploited for quantification of autophagosome biogenesis, but given the multistep nature of this pathway, it is also important to consider the autophagic degradative activity [80]. The rate at which cargos are recognized, segregated and degraded through the autophagy pathway is defined as autophagic flux and it can be measured using diverse methods reviewed elsewhere [80].

The formation of the phagophore is initiated by heterotypic fusion of vesicles, which are probably derived from the ER and recycling endosomes, although other possible membrane sources like the plasma membrane and mitochondria could also be involved [81–83]. The ULK kinase complex is the first functional cluster of proteins assembling at the site of phagophore nucleation, and it is formed by the Unc-51 like autophagy activating kinase 1 or 2 (ULK1/2) and the regulatory subunits ATG13, ATG101 and focal adhesion kinase family interacting protein of 200 kD (FIP200) [84]. This complex regulates phagophore biogenesis and is modulated by mTORC1, which is in turn governed by a variety of

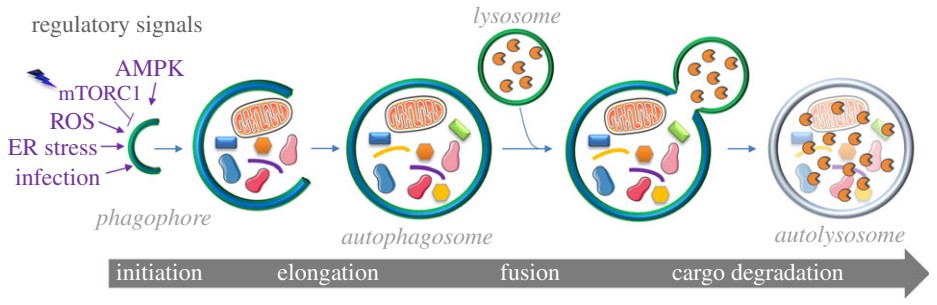

royalsocietypublishing.org/journal/rsob    Open Biol. 9: 190009

**Figure 3.** Schematic representation of the key steps of the autophagy process. Autophagy initiation is under the control of several regulatory signals such as ER stress, ROS production, AMPK or mTORC1 signalling, and the presence of microorganisms. Autophagy begins with the formation of a small cistern, the phagophore, which elongates and sequesters cytoplasmic components such as protein aggregates and organelles. Closure of the phagophore generates a double-membrane vesicle-denominated autophagosome. Subsequent fusion of the autophagosome with lysosomes results in the formation of autolysosomes, where lysosomal hydrolases degrade the cargo contained in the interior of these vesicles (*see text for details*).

upstream signals including growth factors and nutrients such as amino acids and glucose [85] (figure 3). mTORC1 represses autophagy through direct phosphorylation of ULK1 and ATG13; and the absence of the aforementioned signals triggers autophagy initiation [72]. ULK1 is also positively regulated by AMPK, which senses the cellular energy status and is activated when intracellular AMP increases, reflecting a decrease in the availability of ATP [86]. Once autophagy is initiated, the autophagy-specific class III phosphatidylinositol 3-kinase (PI3 K-III) complex, which is composed by the phosphatidylinositol 3-kinase VPS34, Beclin-1 (BECN1), p150 and ATG14 L along with AMBRA1, associates at the sites of phagophore nucleation [87]. This complex is responsible for the local synthesis of phosphatidylinositol-3-phosphate (PI3P), a lipid that is important for the subsequent steps of autophagosome biogenesis [87]. The last functional cluster of proteins that seems to be essential in the early phases of phagophore nucleation is comprised by the transmembrane protein ATG9A and its trafficking machinery, which includes some of the PtdIns3P binding proteins, WD-repeat protein interacting with phosphoinositides 1–4 (WIPI1–4) and ATG2A/2B [88].

Once the phagophore is formed, the PI3P generated by the PI3 K-III complex promotes the association of additional PI3P-binding proteins, which facilitate the recruitment of additional factors that oligomerize into functional complexes that participate in the elongation step (table 1) [89]. Recruited proteins include components of the two ubiquitin-like conjugation systems, which promote both phagophore elongation and closure [90]. The first ubiquitin-like conjugation system leads to the formation of an oligomer constituted by the ATG12–ATG5 conjugate, and ATG16L1, which is tightly associated with the expanding phagophore [63]. The formation of this complex is mediated by the E1 enzyme ATG7 and the E2 enzyme ATG10, which covalently link the ubiquitin-like ATG12 to ATG5, and the resulting ATG12–ATG5 conjugate interacts with ATG16L1 [68]. The second ubiquitin-like conjugation system leads to the N-terminal lipidation of the members of microtubule-associated protein 1A light chain 3 (LC3) protein family, which is composed in humans by LC3A, LC3B, LC3C and the Gamma-aminobutyric acid receptor-associated protein (GABARAP), GABARAPL1 and GABARAPL2 [91]. Members of the ATG4 cysteine protease family (i.e. ATG4A to ATG4D) cleave LC3 proteins at the C-terminal to expose a glycine residue, producing LC3-I [92]. Upon autophagy induction, cytoplasmic LC3-I is activated by ATG7 and, via the E2 enzyme ATG3, is conjugated to the amino group of

phosphatidylethanolamine (PE) molecules present in the phagophore membrane, to produce LC3-II [93]. This later step is guided by the ATG12–ATG5–ATG16L1 oligomer, which enhances both the E2 enzyme activity of ATG3 and recruits it to the forming autophagosome [94]. Once the expansion of the autophagosome is completed, most of the components of the ATG machinery dissociate from the surface of these vesicles and relocate to the cytoplasm, where they can be re-used. During selective autophagy, autophagy receptors, such as Sequestosome-1 (p62/SQSTM1), next to BRCA1 gene 1 protein (NBR1), calcium-binding and coiled-coil domain-containing protein 2 (NDP52), Optineurin (OPTN), FUN14 domain-containing protein 1 (FUNDC1) and BCL2/adenovirus E1B 19 kDa protein-interacting protein 3 (BNIP3); direct specific cargos to autophagosomes via their LC3-interacting-region domains, which mediate the interaction with the LC3 protein pool in the internal autophagosomal surface [95].

Subsequently, a series of coordinated events mediate fusion of autophagosomes with lysosomes, to generate autolysosomes, the final compartments where degradation of the cargo takes place. However, autophagosomes must first fuse with early and/or late endosomes to form organelles known as amphisomes, prior to fusion with lysosomes [96]. Factors associated with the formation of autolysosomes include motor proteins from the dynein, kinesin and myosin protein families, which facilitate the movement of the autophagosomes along the microtubules and actin filaments towards the perinuclear region of the cell, where lysosomes usually concentrate [97,98]. Fusion of autophagosomes with lysosomes requires tethering, which involves the activation of the GTPase RAB7 by the CCZ1-MON1A complex, and its subsequent interaction with the homotypic fusion and vacuole protein sorting (HOPS) complex. The HOPS complex is required to engage soluble *N*-ethylmaleimide-sensitive fusion attachment protein receptor (SNARE) proteins, including syntaxin 17 (STX17), vesicle-associated membrane protein 8 (VAMP8), synaptosomal-associated protein 29 (SNAP-29) and the synaptobrevin homologue YKT6 [99,100]. Additional factors, which sometimes are tissue-specific, participate in the regulation of autophagosome–lysosome fusion [99].

The autophagosomal membrane and the cargo are broken down inside autolysosomes by lysosomal hydrolases such as cathepsin B, L and D. As LC3-II is also incorporated on the internal surface of the autophagosomes, part of this lipidated protein remains trapped in the interior of autolysosomes and therefore is degraded together with the cargo [101,102].

royalsocietypublishing.org/journal/rsob    Open Biol. 9: 190009

# 2. Role of autophagy during arboviral infections

## 2.1. Preface

Viruses depend on and exploit the host-cell machinery for progeny production, thereby modulating and hijacking multiple cellular pathways. In this review, we will summarize key concepts related to the induction and regulation of autophagy over the course of DENV, WNV, ZIKV and CHIKV infections and delineate how this pathway may control the outcome of the infection. We will focus on macroautophagy, as microautophagy and chaperone-mediated autophagy have not yet been studied in the context of these viruses.

There are multiple contradictory results and conclusions in the literature. Although some of these discrepancies could be due to differences in the virus strains and cell lines used for the experiments, others are probably linked to the way autophagy assays have been performed and interpreted. Researchers often examine the steady-state levels of autophagy marker proteins like LC3 or p62, but this does not provide information on the dynamics of this process, the autophagic flux. An increase in the steady-state levels of LC3-II can indicate either induction (i.e. more of this conjugate is produced) or inhibition (i.e. there is no turnover of LC3-II in the lysosomes). Analogously, a decrease in LC3-II levels can also indicate either induction (i.e. LC3-II is rapidly degraded) or inhibition (i.e. LC3-I fails to be converted into LC3-II). Moreover, most of the compounds currently used to inhibit or activate autophagy, like 3-methyladenine (3-MA), wortmannin or rapamycin, are not specific, and consequently, eventual effects on the virus life cycle could be indirect and not linked to a change in the autophagic flux. Similarly, numerous recent discoveries have revealed that ATG proteins are involved in other cellular pathways [103–105] and consequently the depletion of only one of them is not sufficient to conclude that autophagy is involved in a specific aspect of the virus replication cycle. In this regard, it is important to note that there are the so-called non-conventional types of autophagy, which do not require the entire ATG machinery. As a result, the depletion of a single ATG protein does not always guarantee the block of autophagy. Finally, it is important to also keep in mind that LC3-positive puncta, which is often used as a method to assess autophagy induction, do not always represent autophagosomes [106–109].

Thus, the objective of this review is to summarize the literature on the interaction of autophagy and DENV, WNV, ZIKV and CHIKV, and to highlight the experimental approaches to allow the reader to have a critical evaluation of the currently available evidences.

## 2.2. Dengue virus

### 2.2.1. Autophagy induction and autophagic flux during infection

The induction of autophagy during DENV infection has been observed in numerous mammalian cell lines, including Huh7, HepG2, U937, HUVEC, HEK293, HeLa, BHK-21, Vero and Madin-Darby canine kidney (MDCK) cells, by analysing the presence of autophagosomes [110–117]. The first report, by Lee and colleagues, showed the induction of GFP–LC3 puncta formation in DENV-infected Huh7 cells in a multiplicity-of-infection (MOI)-dependent manner [110]. These observations were soon corroborated by others in diverse cellular models and using different methods [111–113]. For example, LC3 puncta accumulation was shown to correlate with LC3 lipidation (i.e. LC3-II synthesis), as assessed by western blot [110,112,114]. Similarly, enhanced LC3-I conversion into LC3-II and an increased number of autophagy-like vesicles were observed at 24, 36 and 48 h post-infection (hpi) by western blot and electron microscopy, respectively, in endothelial HUVECs and EA.hy926 cell models [117]. Moreover, DENV-induced autophagosome biogenesis was shown to be decreased by the PI3 K-III inhibitors 3-MA and wortmannin, further supporting the notion that DENV infection induces autophagy [110,112]. As UV-inactivated DENV is unable to induce LC3-positive puncta formation and bona fide autophagosomes in infected cells, this observation indicated that active viral replication triggers autophagy [110]. Interestingly, ectopic expression of the DENV NS4A protein was observed to induce LC3 puncta formation and LC3 lipidation in HeLa cells, suggesting that NS4A may trigger the putative autophagic response [112]. Similarly, treatment of HMEC-1 endothelial cells with DENV NS1 protein induced p62 degradation, LC3-I to LC3-II conversion and the presence of LC3 puncta as assessed by western blot and immunofluorescence staining [118]. It is interesting to note that DENV NS1 increased the permeability of these cells and vascular leakage in BALB/c mice, a phenomenon that was demonstrated to be dependent on autophagy [118]. Additional in vivo models and studies involving primary cell culture have helped to confirm some of the described in vitro observations. For example, DENV also induced the formation of autophagosomes in primary human monocytes, which are considered important targets during DENV infection [110]. Additionally, brains of suckling mice infected with DENV showed an induction of endogenous LC3-positive puncta formation at 5 days post-infection (dpi) [114]. In addition, DENV-infected animals displayed a reduction in the p62 levels and induction of LC3-II at 3 and 5 dpi, further demonstrating an autophagy induction [114]. Altogether, these studies indicate that autophagosome formation is initiated upon DENV infection, possibly via NS4A and NS1 expression and may depend on the autophagy PI3 K-III complex.

Autolysosome formation and increase in autophagic flux upon DENV infection have been observed in several studies [110,111], though one investigation reached an opposite conclusion [113]. Treatment of DENV-infected Huh7 cells with vinblastine, a microtubule disrupting agent that also inhibits autophagosome–lysosome fusion, enhanced LC3-II levels when compared to untreated-infected cells, as assessed by immunoblotting [110]. Moreover, co-localization of LC3 puncta with the lysosomal marker LAMP1 and the Lysotracker dye was observed in Huh7 and Huh7.5 cells at 24 and 36 hpi with DENV, which was suggested to indicate an enhancement of the autophagic flux [110,111]. In another study, however, LC3 puncta did not co-localize with LAMP2 at 36 hpi in DENV-infected Huh7 cells, although autophagy was induced as assessed by measuring the steady-state and flux levels of LC3-positive vesicles by quantitative image-based flow cytometry [113]. Moreover, bafilomycin A1, an inhibitor of autolysosome acidification and hence cargo degradation, did not lead to an increase in GFP–LC3 puncta accumulation in DENV-infected cells, suggesting an impairment in autophagic flux. In contrast to the above studies, the authors concluded that DENV activates

autophagosome formation but inhibits the autophagic flux [113]. It is difficult to determine why very similar studies obtained different results. This discrepancy may be related to the method employed to evaluate the autophagic flux (confocal microscopy versus image-based flow cytometry versus western blot), the evaluated time points or the compound used to inhibit lysosomal degradation (vinblastine versus bafilomycin A1). Nonetheless, it remains to be firmly established whether DENV infection induces or blocks autophagic flux. The use of alternative assays like the one based on the RFP–GFP–LC3 tandem construct [119] could be of help in solving this issue.

### 2.2.2. Is autophagy induction beneficial or detrimental for DENV replication?

Although it is clear that autophagy is induced in DENV-infected cells, the role of this pathway in the replication of the virus is, however, more intricate and complex. Contrasting results have been published which will be addressed below (table 2).

#### 2.2.2.1. Evidence pointing towards a beneficial effect of autophagy in DENV infection

Lee and co-workers found that infectious virus particle production was significantly decreased in $Atg5^{-/-}$ knockout mouse embryonic fibroblasts (MEFs) compared to the control, thereby suggesting that an intact autophagy pathway promotes viral replication and release [110]. Other lines of evidence indicating a proviral role of autophagy in DENV replication include studies using drug inhibitors and/or siRNAs that target different steps of autophagy and functional groups of ATG proteins. For example, treatment with 3-MA or wortmannin, and ATG12 or BECN1 siRNA-based silencing, were found to decrease viral replication and infectious viral titres in diverse cell types [110–112,124,128]. Conversely, treatment with rapamycin, a potent stimulator of autophagy through mTORC1 inhibition, increased DENV infectious particle production in a dose-dependent manner [110,124]. In a recent study, however, it has been shown that DENV infection and egression are unaltered in ULK1, BECN1 or ATG5 knockout HeLa cells, whereas viral replication was impaired in cells lacking ATG9, LC3B or VPS34, knockout cells suggesting that this virus only exploits specific autophagy components [131]. Similar to the results observed in vitro, treatment of suckling mice with rapamycin promoted viral replication as shown by an increase in DENV titres, which correlated with a more severe clinical outcome and a reduction in the survival rate of the mice [114]. However, it is worth mentioning that rapamycin is a known immune-suppressor [141,142], and therefore, these results need to be carefully interpreted. Additionally, it was also shown that the treatment of suckling mice with 3-MA, which inhibited LC3-II synthesis and p62 degradation, improved the survival rate of the mice and their clinical scores [114]. In a different murine model, the use of SP600125 to inhibit JNK activation in mice infected with DENV, reduced LC3-II levels, viral titres, disease symptoms and prolonged the survival rate of the infected mice [127]. Collectively, the above evidence suggests that autophagy, or at least specific autophagy components, are required for successful DENV infection in mammalian cell lines and probably also in vivo. Of note,

these same components are also involved in other pathways, like the recruitment of LC3 onto endosomes, for example, during LC3-mediated phagocytosis [109].

Several studies have focused on the possible mechanisms by which autophagy is beneficial for the virus. Early observations suggested that DENV RNA replication occurs within autophagy-associated vesicles, but other studies have challenged this view. In HepG2 cells, components of the translation/replication machinery like NS1 and double-stranded RNA (dsRNA), which marks active sites of viral replication, were found to co-localize with marker proteins that label amphisomes, such as mannose-6-phosphate receptor, LAMP1 and LC3 [124,143]. Partial co-localization of NS1 and LC3 puncta was also detected in brain tissues from DENV-infected mice [114]. By contrast, no detectable co-localization between LC3 and the viral proteins NS1 and NS3, or dsRNA, was observed in DENV-infected Huh7.5 cells [111]. Comparable results were published for DENV-infected monocytic U937 cells [116]. Moreover, recent high-resolution electron microscopy studies revealed that DENV replication complexes assemble in extensive ER-associated membrane rearrangements that form in an autophagy-independent manner [40,144]. Thus, even if co-localization between viral components and autophagy marker proteins was observed in early studies, it is unlikely that DENV RNA replication occurs in association with autophagosomes or autophagy-derived vesicles.

Other studies evaluated the replication cycle of the virus to delineate the step where autophagy promotes viral replication. Heaton and co-workers revealed that neither entry nor viral protein translation is affected in DENV-infected Huh7.5 cells treated with 3-MA or siRNA targeting BECN1 [111]. In another study, autophagy was suppressed in BHK-21 cells by spautin-1, an unspecific inhibitor of BECN1 de-ubiquitination, and this severely hampered the specific infectivity of progeny viral particles [115]. Stimulation of autophagy by nicardipine, (a modulator of intracellular $Ca^{2+}$ flux) or by rapamycin was found to have the opposite effect (i.e. the specific infectivity of progeny virions was increased), suggesting a possible effect of autophagy on viral maturation [121]. $Ca^{2+}$ levels, however, probably influence numerous other cellular processes, such as the enzymatic activity of furin [145], which is required in the latest stages of viral maturation in the TGN. In agreement with these observations, intracellular DENV RNA levels in ATG9, LC3B and VPS34 knockout HeLa cells are similar to the control during the first 24 hpi; but they decrease at later time points, suggesting an effect posterior to viral RNA replication processes and possibly first replication cycle [131]. Finally, DENV cell-to-cell spread has been linked to specific components of the autophagy machinery [126]. Extracellular vesicles released from DENV-infected Huh7 cells were reported to contain LC3-II, DENV proteins (E, NS1, prM/M) and infectious viral RNA [126]. Interestingly, these vesicles were also detected in the serum of a DENV-infected patient [126]. The relevance of these LC3-II-containing vesicles for viral spread in the context of human infection is still unknown.

Next to this, three independent studies proposed that the virus benefits from the induction of lipophagy, the selective autophagy of lipid droplets [111,120,128]. Lipophagy is activated during DENV infection, thereby increasing the β-oxidation rates and consequently ATP levels, which promotes replication [111]. This phenomenon has been studied in Huh7, Huh7.5 and HepG2 cells, where DENV infection

royalsocietypublishing.org/journal/rsob Open Biol. 9: 190009

**Table 2.** Summary of the literature describing an antiviral or proviral role of autophagy or ATG proteins over the course of specific flavivirus infections.

| (a) DENV | | | |
| --- | --- | --- | --- |
| (i) (*in vitro*) | | | |
| **cell type** | **experimental approach** | **role of autophagy**[a] | **references** |
| WT MEFs | rapamycin treatment | proviral | [110] |
| | 3-MA treatment | | |
| *Atg5*$^{-/-}$ MEFs | — | proviral | |
| Huh7 | rapamycin treatment | proviral | |
| | 3-MA treatment | | |
| Huh7 | stable p62 overexpression | antiviral (p62) | [113] |
| Huh7 | 3-MA treatment | proviral | [111] |
| Huh7.5 | ATG12 and BECN1 siRNA | | |
| BHK | | | |
| HepG2 | | | |
| HepG2 | AMPK$\alpha$ siRNA (used as an inhibitor of lipophagy during DENV infection) | proviral | [120] |
| | compound C treatment (AMPK inhibitor) | | |
| BHK-21 | spautin-1 treatment | proviral (supports viral maturation) | [121] |
| | rapamycin treatment | | |
| | nicardipine treatment | | |
| | 3-MA treatment | | |
| MDCK | Wortmannin treatment | proviral | [112] |
| | 3-MA treatment | | |
| KU812 | Atg4B$^{C74A}$ overexpression | proviral | [122] |
| A549 | rapamycin treatment | no effect | [123] |
| | 3-MA treatment | | |
| | BECN1 and ATG7 siRNA | | |
| THP-1 | rapamycin treatment | antiviral | |
| | 3-MA treatment | | |
| | BECN1 and ATG7 siRNA | | |
| U937 (ADE conditions) | rapamycin treatment | antiviral (mild effect) | [116] |
| | L-asparagine treatment | | |
| | Vps34$^{dn}$ overexpression | | |
| HepG2 (DENV-2) | rapamycin treatment | proviral | [124] |
| | 3-MA treatment | | |
| | L-asparagine treatment | | |
| HepG2 (DENV-3) | rapamycin treatment | proviral | [125] |
| | 3-MA treatment | | |
| | L-asparagine treatment | | |
| Huh7 | ATG5 and ATG9 siRNA | proviral (autophagy participates in virus spread in co-cultured cells) | [126] |
| *Atg5*$^{-/-}$ MEFs | — | | |
| *Atg5*$^{-/-}$ MEFs | — | proviral | [127] |
| Huh7 | IRE1$\alpha$ inhibitor | proviral (indirect) | [127] |
| | IRE1$\alpha$ and eIF2$\alpha$ shRNA | | |
| | SP600125 treatment | | |
| HepG2 | AUP1 siRNA | proviral (AUP1) | [128] |

(*Continued.*)

**Table 2.** (*Continued.*)

| cell type | experimental approach | role of autophagy[a] | references |
|---|---|---|---|
| K562 (ADE conditions) | rapamycin treatment | proviral | [129] |
| | 3-MA treatment | | |
| | CRISPR-Cas9 knockout of ATG5 | | |
| HBMEC | FAM134B siRNA | antiviral (FAM134B, reticulophagy) | [130] |
| HeLa | CRISPR-Cas9 knockout of ATG9A, VPS34 and LC3B | proviral | [131] |

(ii) (*in vivo*)

| animal model | experimental approach | role of autophagy[a] | references |
|---|---|---|---|
| suckling mice | rapamycin treatment | proviral | [114] |
| | 3-MA treatment | no effect | |
| suckling mice | SP600125 treatment | proviral | [127] |

## (b) WNV

(i) (*in vitro*)

| cell type | experimental approach | role of autophagy[a] | references |
|---|---|---|---|
| MCCs | trehalose treatment | no effect | [132] |
| BSCs | wortmannin treatment | no effect | |
| | 3-MA treatment | proviral (unspecific effect) | |
| BHK | ATG5 shRNA | no effect | |
| $Atg5^{-/-}$ MEFs | — | no effect | |
| $Atg5^{-/-}$ MEFs | — | antiviral (only at low infectious dose, MOI 0.01) | [133] |
| HeLa | TAT-BECN1 peptide | antiviral | |
| $Atg5^{-/-}$ MEFs (m5−7 clone, suppresses Atg5 expression upon doxycycline treatment) | — | no effect | [134] |
| HEK293T | ATG7 siRNA | no effect | |
| HeLa | TAT-BECN1 peptide | antiviral | [135] |
| Vero | 3-MA treatment | proviral (WNV-B13) | [136] |
| | | no effect (WNV-NY99) | |

## (c) ZIKV

(i) (*in vitro*)

| cell type | experimental approach | role of autophagy[a] | references |
|---|---|---|---|
| primary skin fibroblasts | torin1 treatment | proviral | [137] |
| | 3-MA treatment | | |
| fNSCs | rapamycin treatment | proviral | [138] |
| | 3-MA treatment | | |
| | CQ treatment | | |
| | ATG3 and ATG13 siRNA | | |
| HeLa | rapamycin treatment | proviral | |
| | 3-MA treatment | | |
| | CQ treatment | | |
| $Atg3-/-$ and $Atg5-/-$ MEFs | — | proviral | |
| HUVEC | wortmannin treatment | proviral (although NS3 protein is increased by wortmannin and decreased by BECN-1 shRNA) | [139] |
| | CQ treatment | | |
| | BECN1 shRNA | | |
| | rapamycin treatment | no effect | |

(*Continued.*)

**Table 2.** (Continued.)

| | | | |
|---|---|---|---|
| JEG-3 | rapamycin treatment | proviral | [140] |
| | Torin1 treatment | | |
| | 3-MA treatment | | |
| | CQ treatment | | |
| | Bafilomycin A1 treatment | | |
| HBMEC | FAM134B siRNA | antiviral (FAM134B, reticulophagy) | [130] |
| HeLa | CRISPR-Cas9 knockout of ULK1, BECN1, ATG9A, VPS34 and LC3B | proviral | [131] |
| (ii) (in vivo) | | | |
| **animal model** | **experimental approach** | **role of autophagy**[a] | **references** |
| Atg16l1[HM] mice | — | proviral | [140] |
| WT C57BL6 mice | HCQ treatment | proviral | |

[a]Measured by assessing viral titres, percentage of infection, extracellular or intracellular RNA.

leads to a decrease in the size of lipid droplets and free fatty acid levels at 48 hpi, which correlates with an increase in LC3 puncta formation [111,128]. Furthermore, lipid droplets were found to co-localize with autophagosomal and lysosomal marker proteins to a higher extent in cells exposed to DENV than in mock-treated cells [111]. In line with these observations, 3-MA treatment and silencing of ATG12 or BECN1 to inhibit autophagy, restored lipid droplet mass and decreased co-localization of LC3 with lipid droplets [111]. A subsequent study demonstrated that DENV-induced lipophagy, but not basal autophagy, depends on AMPK kinase activity and inhibition of mTORC1 signalling [120]. Moreover, it was recently found that during DENV infection of HepG2 cells, NS4A and NS4B viral proteins interact with Ancient ubiquitous protein 1 (AUP1), a lipid droplet-localized membrane protein [128]. This interaction drives the relocation of AUP1 from lipid droplets to autophagosomes, triggering lipophagy [128]. Furthermore, deletion of AUP1 arrests DENV-induced lipophagy and impairs viral production [128]. Altogether, these studies underscore the importance of lipophagy for DENV replication cycle and highlight the role that NS4A and NS4B play to hijack this pathway.

Additional components of the autophagy machinery have also shown to play a role during DENV infection, specifically in conditions of antibody-dependent enhancement (ADE). It is generally accepted that pre-existing, cross-reactive, poorly neutralizing antibodies can enhance DENV infectivity and replication in Fcγ receptor-expressing cells, such as macrophages and monocytes, a process that eventually leads to vascular leakage [146]. Treatment of K562 myelogenous leukaemia cells with rapamycin prior to infection with DENV-antibody-enhancing complexes, increased intracellular viral RNA levels at 48 hpi [129]. Conversely, 3-MA treatment reduced intracellular DENV RNA after infection. Furthermore, in ATG5 knockout K562 cells, a decrease in intracellular viral RNA synthesis was detected [129]. Moreover, DENV–antibody complexes led to an increase in ATG12–ATG5 conjugate levels in monocytic THP-1 cells, and ATG12 and ATG5 transcripts and ATG5 and LC3-II protein levels in K562 cells [129,147]. This led to a negative regulation of retinoic acid-inducible gene I (RIG-I)

and the melanoma differentiation-associated protein 5 (MDA-5) signalling pathways, which in turn dampened interferon type I (IFN-I) response and promoted viral replication in THP1 cells [147,148]. Moreover, in K562 cells, overexpression of ATG5 impaired NF-κB activation, which eventually led to increased DENV RNA [129]. In addition, in an independent study, infection of pre-basophil-like KU812 and immature mast-like HMC-1 cell lines with DENV in the presence of cross-reactive enhancing antibodies was shown to induce autophagy, and inhibition of this pathway through the generation of a KU812 stably expressing the mutant Atg4B[C74A] reduced viral replication [122].

### 2.2.2.2. Evidence pointing towards an antiviral role of autophagy in DENV infection

A few studies have suggested that autophagy may act as an antiviral pathway in DENV infection, but evidences are less compelling than those indicating that autophagy is proviral (table 2). For example, induction of autophagy by rapamycin in U937 monocytic cells resulted in a decrease in extracellular virus output, whereas downregulation of autophagy by L-asparagine had no effect in DENV infectious particle production [116]. This finding was confirmed by another study in which autophagy induction by rapamycin in monocytic THP-1 cells significantly decreased the progeny DENV titre, while 3-MA, or siRNA targeting BECN1 or ATG7-mediated inhibition of autophagy increased the viral titre [123]. Studies by the same authors additionally identified miR-146a as a regulator of both autophagy and innate immune responses during DENV infection [123,149]. It was initially shown that expression of miR-146a facilitates DENV replication by targeting TRAF6, an essential innate immune signalling adaptor that activates the nuclear factor kappa-light-chain-enhancer of activated B cells (NF-κB) transcription factor and the production of IFN-I [149]. In addition, miR-146a-mediated TRAF6 downregulation blocked DENV-induced autophagy in THP-1 cells [123]. Furthermore, silencing of ATG7 or BECN1 by siRNA transfection in DENV-infected cells decreased the production of proinflammatory cytokines, confirming a possible role of

autophagy in modulating DENV-induced immune response in monocytic cells [123].

Furthermore, the autophagy receptor p62 was described to directly hamper DENV replication [113]. Indeed, reduced DENV replication was observed in Huh7 cells stably overexpressing p62 [113]. Interestingly, DENV was found to counteract p62 expression, as the expression level of p62 was shown to be reduced during infection, even when the autophagic flux was progressively blocked with bafilomycin A1. Moreover, DENV-induced p62 reduction was abolished by treatment with the proteasomal inhibitor epoxomycin, suggesting that DENV induces p62 proteosomal degradation to subvert an autophagy-mediated antiviral response [113]. Of note, these data have to be pondered in the light of the fact that p62 also represents a hub to coordinate autophagy and oxidative stress (e.g. [150,151]).

### 2.2.3. Autophagy crosstalk with other cellular stress pathways

McLean and co-workers found that DENV infection of HEK293T, HeLa, Vero, MDCK cells and MEFs prevents cell death caused by several stimuli, including DNA damage through camptothecin treatment, inhibition of kinases induced with staurosporine and protein synthesis inhibition by cyclohexamide [112]. This cytoprotective effect was abolished in $ATG5^{-/-}$ MDCK cells, suggesting that autophagy stimulation during DENV infection is a prosurvival and proviral mechanism [112]. In agreement with the aforementioned observation, 3-MA treatment further reduced the number of surviving DENV-infected cells, indicating that autophagy may contribute to cell survival under these conditions [117]. Indeed, in the same study, autophagy inhibition was found to upregulate apoptosis in HUVECs and EA.hy926 cells [117]. Finally, it was identified that NS4A overexpression was sufficient to confer protection from cell death induced by camptothecin or staurosporine treatment in MDCK cells, and this protection was associated with the induction of autophagy [112].

Additional evidence indicates that DENV also activates the protein kinase RNA-like endoplasmic reticulum kinase (PERK) branch of the UPR at early time points after infection in MDCK and Huh7 cells and in MEFs, which can trigger autophagy and ROS production, and through a positive-feedback loop further stimulates autophagy [115,127]. In DENV-infected Huh7 cells, PERK signalling leads to eIF2$\alpha$ phosphorylation, thereby enhancing the translation of ATF4 and ultimately upregulating the expression of ATG proteins (e.g. ATG12) and autophagy at 12 and 24 hpi [127]. The inositol requiring kinase 1(IRE1$\alpha$) and c-Jun N-terminal kinase (JNK) branch of the UPR is required for this autophagy stimulation [127]. IRE1$\alpha$–JNK is a major signalling pathway that induces BCL2 phosphorylation and causes dissociation of the BECN1–BCL2 complex to release BECN1 and thereby promotes autophagy [152]. Indeed, blocking JNK activation using the specific inhibitor SP600125 in DENV-infected mice, reduced DENV-induced autophagy [127].

Collectively, these studies suggest a mutually exclusive relationship between autophagy and apoptosis, and indicate that the UPR and the JNK signalling pathways are important regulators of autophagy during DENV infection. However, they also evoke the possibility that autophagy is not directly subverted by DENV but it could rather represent a prosurvival response of the cell to adapt to stress.

## 2.3. West Nile virus

Autophagy induction during WNV infection has been found to be cell-type specific. For example, infection of Vero cells stably expressing GFP–LC3 with the highly pathogenic strain New York 99 (WNV-NY99) led to more pronounced steady-state levels of LC3 puncta and LC3 lipidation at 24 hpi when compared to mock-infected control cells, although no effects on p62 levels were observed [132]. Similarly, an increase in steady-state amounts of LC3 puncta and lipidated LC3 were seen in SK-N-SH, a neuroblastoma cell line, infected with WNV-NY99 but at an earlier time point (6 hpi) [133]. Furthermore, SK-N-SH cells treated with the lysosomal protease inhibitors E64d and pepstatin A had increased LC3 lipidation following WNV-NY99 infection, indicating that WNV indeed enhances autophagic flux [133]. In agreement with this observation, Vero cells exposed to WNV-NY99 and treated with bafilomycin A1 or chloroquine (CQ), also showed more LC3-II accumulation when compared to the control cells [132]. Moreover and similarly to what has been observed for DENV, 3-MA was found to inhibit LC3-positive puncta formation in WNV-infected Vero cells [132]. Similarly, ATG5 knockdown in BHK-21 cells using shRNA also reduced LC3-positive puncta formation in WNV-infected cells [132]. Lastly, increased steady-state LC3 lipidation was detected at 24 and 48 hpi with WNV-NY99 in a three-dimensional CNS model [132]. By contrast, no enhanced steady-state LC3 lipidation was detected following infection with WNV-NY99 in HEK293T, Huh7, Huh7.5, A549 cells and human skin fibroblasts (HFF) [134]. Collectively, these observations suggest that the induction of autophagy following WNV infection is cell-type specific. Alternatively, the discrepancies can be explained by differences between the studies (e.g. the MOI used, the analysed time points, inherent susceptibility of the cells to infection, the metabolic cell status and the autophagy measurement method). Therefore, a comparative study should be performed using cell lines that have previously shown contradictory results, to be able to pinpoint whether this pathway is induced by WNV infection. The virulence of the WNV strain, however, does not appear to be a determinant factor as a direct comparison of WNV-NY99 and a low virulent Kenyan WNV isolate appeared to equally stimulate autophagy in Vero cells, as assessed through the analysis of steady-state levels of LC3B-II at 24 hpi by immunoblotting [132]. Infection of Vero cells expressing GFP–LC3 with multiple variants of WNV (B13, ArD27875, Egypt101 and B956), however, resulted in GFP–LC3 puncta accumulation and in increased steady-state levels of lipidated LC3, as assessed by western blot at 24 and 48 hpi. In comparison, compared to the uninfected control, WNV-NY99-infected Vero cells did not redistribute the GFP–LC3 signal and the steady-state levels of LC3-II did not change [136]. The authors of this study suggested that mutations in the NS4A and NS4B of WNV could be responsible for the discrepancies but further *in vitro* and *in vivo* characterization of these mutants is required to eventually understand the molecular mechanisms behind these discrepancies in autophagy regulation between WNV strains [136,153].

The role of autophagy in promoting or restricting WNV replication is also a controversial subject (table 2). While some reports indicate that autophagy modulation does not affect the virus replication [132–134], other studies suggest otherwise [133,135]. No differences in viral titres were observed in WNV-NY99-infected HEK293T cells transfected with siRNA

royalsocietypublishing.org/journal/rsob    Open Biol. 9: 190009

targeting ATG7 compared to the cells treated with an siRNA control [134]. In line with this result, treatment of primary mouse cortical cultures (MCCs) with trehalose, an mTORC1-independent inducer of autophagy, did not have a significant effect on progeny WNV infectious titres at 72 h when infected with an MOI of 3 [132]. In addition, inhibition of Atg5 with shRNA had no effect on WNV infectious particle production from 6 to 24 hpi as compared to cells transduced with a shRNA control [132]. On the other hand, PI3 K-III inhibitors 3-MA and wortmannin significantly reduced viral titres of WNV-NY99 in organotypic brain slice cultures at 72 hpi, although this was suggested to be related to pleiotropic effects of these compounds [132]. Similarly, Vero cells treated with 3-MA released less WNV-B13 infectious virus particles, though no effect was observed for WNV-NY99, indicating a strain-specific effect [136]. On the contrary, another study showed a significant enhancement in progeny virus particle production at 24 and 48 hpi but not at 72 hpi following infection with WNV-NY99 at MOI 0.01 in $Atg5^{-/-}$ MEFs when compared to $Atg5^{+/+}$ MEFs or $Atg5^{-/-}$ cells back-transfected with $Atg5$ [133]. Additionally, they showed that $Atg5^{-/-}$ MEFs had higher levels of WNV genomic RNA than parental cells at 6 hpi, as measured by RT-qPCR. No effect was, however, seen at higher MOIs (e.g. MOI 0.1 and 1) [133]. Interestingly, treatment of HeLa cells with TAT-BECN1, a potent autophagy-inducing peptide, decreased WNV titres without affecting viral entry or cell survival [133,135]. Similarly, treatment of WNV-infected (strain Egypt 101) neonatal mice with the TAT-BECN1 peptide, led to a pronounced reduction in brain viral titres, clinical paralysis and mortality caused by the virus [135]. Overall, these data suggest that even though autophagy is not required for WNV replication, strong induction of this pathway could have a detrimental effect on the virus, possibly due to non-specific degradation of viral components or other antiviral pathways modulated by autophagy.

WNV-NY99 infection leads to ER stress and UPR induction, though this has been mainly associated with the initiation of apoptosis in SK-N-MC neuroblastoma cells rather than autophagy [154]. Furthermore, UPR activation appears to be strain-specific and cell-type-dependent, as the attenuated Kunjin WNV subtype only activates PERK-mediated translation and CHOP transcription in Vero cells, whereas WNV-NY99 was described to upregulate all three pathways of the UPR (PERK, IRE1$\alpha$ and ATF6) in SK-N-MC cells [154,155]. Moreover, comparable levels of spliced XBP1 were observed for different WNV strains in Vero cells, which is indicative of similar UPR induction, yet not all of these strains were found to induce autophagy [136]. Together, this shows that the importance of the cellular stress response in autophagy induction is not exactly clear and future studies are required to address this question.

## 2.4. Zika virus

The interaction between ZIKV and autophagy has only recently been described. Electron micrographs of primary skin human fibroblasts infected with a clinical ZIKV isolate from French Polynesia showed the presence of double-membrane vesicles resembling autophagosomes at 72 hpi [137]. Enhanced autophagosome formation has also been observed in HeLa and HUVEC cell lines, and in MEFs, which displayed increased LC3 lipidation and p62 degradation upon ZIKV infection [138,139]. In addition, LC3-I to LC3-II conversion

and LC3 puncta formation were observed at 12 hpi in the human trophoblast cell type JEG-3, following exposure to the Brazilian ZIKV strain Paraiba 2015 in the presence and absence of bafilomycin A1 [140]. Furthermore, accumulation of LC3-positive puncta co-localizing with the viral E protein in proximity to the ER was seen in HFF1 cells at 24 hpi [137]. These findings led the authors to hypothesize that autophagosomes are the sites of ZIKV replication [137]. This notion, however, has been challenged by a more recent study in which ZIKV replication factories were described to be tightly linked to ER membrane invaginations surrounded by rearrangements of the host cell cytoskeleton [156].

Multiple studies reported enhanced autophagic flux during ZIKV infection in different in vitro and in vivo systems and independent of the strain used [137–140]. HUVEC cells transduced with a lentivirus system encoding mTagRFP-mWasabi-LC3, which allows differentiation of autophagosomes (Wasabi+/ RFP+ puncta) from autolysosomes (RFP+ puncta), demonstrated an increase in the autophagic flux from 18 to 24 hpi following infection with ZIKV strain GZ01 [139]. Moreover, due to the association of ZIKV infection with microcephaly [157,158], several efforts have been made to dissect the role of autophagy in ZIKV infection during neuronal differentiation in forming brains. For example, three ZIKV strains of diverse origin (i.e. MR766, IbH30656 and H/PF/2013) were observed to enhance LC3-I conversion into LC3-II in fetal NSCs (fNSCs) when the autophagic flux was monitored in the presence and absence of bafilomycin A1 [138]. Furthermore, steady-state p62 levels were reduced from 6 to 24 hpi, which is consistent with a possible induction of autophagy [138]. Furthermore, an in vivo study using a mouse model for maternal–fetal transmission of ZIKV confirmed the results observed in the previous in vitro studies [140]. Cao and colleagues reported an increase in steady-state levels of LC3-II and a decrease of p62 in the entire placenta of animals infected with ZIKV strain Paraiba 2015 [140]. Lastly, the possible role of virally encoded NS proteins in autophagy induction was investigated in more detail. Lentivirus-based overexpression of ZIKV NS4A or NS4B led to GFP–LC3 puncta accumulation in HeLa cells, and increased LC3 lipidation in fNSC when the autophagic flux was assessed in the presence and the absence of bafilomycin A1. Co-expression of both viral proteins further enhanced the effect, by impairing Akt-mediated positive regulation of mTOR activity [138]. Collectively, the results from these studies show that autophagy could be initiated via NS4A and NS4B by regulating mTOR activation. However, how these proteins interfere with Akt signalling remains to be understood.

Several studies have shown that autophagy induction is beneficial for ZIKV replication and pathogenesis. Treatment of diverse cell types (i.e. JEG-3 cells, primary fibroblasts or fNSCs) with the mTOR inhibitors Torin-1 or rapamycin upon ZIKV infection, resulted in a concomitant increase in autophagosome formation and viral replication, whereas treatment with the inhibitors 3-MA or CQ decreased viral replication [137,138,140]. Similarly, treatment of HUVECs with wortmannin and CQ significantly decreased ZIKV titres [139]. In line with these observations, infection of $Atg3^{-/-}$ and $Atg5^{-/-}$ MEFs with ZIKV reduced virus replication compared to control MEFs [138]. Reduced infectious virus titres were also observed in ATG3- or ATG13-depleted fNSCs, or BECN1-silenced HUVECs [138,139]. In agreement with these studies, knockout of ULK1, ATG9, BECN1, VPS34 or LC3B in HeLa cells, caused a reduction of both ZIKV viral titres and the

royalsocietypublishing.org/journal/rsob Open Biol. 9: 190009

royalsocietypublishing.org/journal/rsob    Open Biol. **9**: 190009

percentage of infected cells at 48 hpi [131]. In addition, treatment of pregnant mice with the autophagy inhibitor hydroxyl-CQ (HCQ), reduced ZIKV titres in the placentas of these mice without influencing the systemic maternal infection [140]. Also, a pronounced decrease in ZIKV titres has been detected in the placentas of virus-inoculated pregnant mice carrying a hypomorphic allele of *Atg16l1*, which was correlated with less pathological damage [140]. Finally, expression of ZIKV NS4A and NS4B in fNSCs caused an impairment in neurosphere formation and differentiation capacity, a phenomenon that was correlated with upregulation of autophagy [138]. Altogether, these studies highlight that ZIKV-induced autophagy has a proviral effect in multiple contexts, hence potentially contributing to the development of the severe clinical manifestations observed in human neonates.

While induction of bulk autophagy is proviral for ZIKV, a recent study suggests that reticulophagy is part of the cellular antiviral response [130]. Downregulation of Reticulophagy regulator 1 (FAM134B), a specific reticulophagy receptor [159], by siRNA-mediated knockdown, boosts intracellular viral RNA levels and infectious titres of both ZIKV and DENV, in human brain microvascular endothelial cells (HBMECs) [130]. The authors of this study showed that FAM134B is cleaved by the flavivirus NS2B3 protease, to facilitate the expansion of the ER rather than its degradation [130]. In addition, NS2B3 is able to disrupt viral protein sequestration in reticulophagy-derived autophagosomes, as demonstrated by the absence of FAM134B- and NS3-positive puncta in U2OS cells co-transfected with DENV NS2B and FAM134B carrying a mutation in the LC3-interacting domain [130]. These results suggest that FAM134B, and in general reticulophagy, may act as a restriction factor for infection. The involvement of other ATG proteins in this context, however, has not been examined and viral proteins were shown to localize with FAM134B but not directly with LC3 [130]. As a result, further studies are required to determine whether FAM134B ablation is indeed impairing reticulophagy during ZIKV infection as other selective autophagy receptors are also known to be involved in this pathway [160]. Nonetheless, an emerging picture could be that while bulk autophagy is beneficial for ZIKV infection, this virus targets FAM134B to assure that the ER and viral proteins are not turned over.

## 2.5. Chikungunya virus

### 2.5.1. Autophagy induction and autophagic flux during infection

Hallmarks of the induction of autophagy during CHIKV infection have been reported for multiple human cell lines, including HEK293, HeLa, HFF and U-87 MG, and also in MEFs and mice using diverse methods [161–163]. Accumulation of GFP–LC3-positive puncta was detected from 4 to 48 hpi following CHIKV infection in MEFs, HEK293, HeLa and human glioblastoma U-87 MG cells expressing GFP–LC3 by fluorescence microscopy and flow cytometry [161–164]. Furthermore, electron microscopy analyses in HeLa and HEK293 cells exposed to CHIKV revealed the accumulation of membranous vesicles reminiscent of autophagosomes [161,163]. In line with these observations, LC3-I conjugation to PE to form LC3-II has been observed in HeLa cells and MEFs between 5 and 24 hpi by western blot analysis [162,163]. Furthermore, a reduction in p62 cellular levels in HeLa cells infected with a clone-derived CHIKV-37997 was observed at 15 hpi [163]. In another study, however, the authors failed to detect accumulation of LC3-II in CHIKV-infected HepG2 cells whereas the same authors did observe enhanced LC3-II accumulation in HeLa cells challenged with CHIKV [165]. This discrepancy is not fully understood yet, and the observed differences between the two cell lines may just reflect differences in the steady-state levels of LC3-II, as the autophagic flux was not investigated. Active replication of CHIKV is required for inducing autophagy, because UV-inactivated virus is unable to enhance LC3-positive puncta formation and LC3-I conversion into LC3-II in MEFs [162]. Although it is not exactly known how CHIKV replication induces autophagy, a study for the closely related Semliki Forest virus (SFV) revealed that accumulation of autophagosomes might depend on the expression of the glycoprotein spike complex [166]. It this case, however, autophagosome accumulation was a result of inhibition of autophagosome degradation rather than an active initiation of an autophagic response [166]. Similar to the *in vitro* studies, higher LC3-II levels have also been observed in a neonatal mouse model for CHIKV at 3 dpi [163]. These data collectively suggest that CHIKV replication may initiate an autophagic response *in vitro* and *in vivo*.

Two independent studies reported co-localization of LC3 puncta with the lysosomal marker protein LAMP1 in HeLa cells infected with CHIKV ECSA genotype or CHIKV-21 strain at MOI 1 and 10 at 24 hpi, which indicates an induction of the autophagic flux that leads to the fusion between autophagosomes and lysosomes [162,165]. Experiments using the tandem construct expressing RFP–GFP–LC3, which allows to distinguish between autophagosomes (i.e. RFP+/GFP+ puncta) from the acidic autolysosomes (i.e. RFP+/GFP− puncta) corroborated these findings [162]. Altogether, these results indicate that autophagy is probably triggered over the course of a CHIKV infection and that it culminates in autolysosome formation.

### 2.5.2. Is autophagy induction beneficial or detrimental for viral replication?

Although early studies on CHIKV and autophagy suggested an antiviral role of this pathway in human and murine cell lines, more recent evidence has revealed proviral effects and the results of these studies will be outlined below (table 3).

#### 2.5.2.1. Evidence pointing towards a beneficial effect of autophagy in CHIKV infection

Multiple studies have highlighted that autophagy induction promotes CHIKV replication in human cells. For example, HEK293 pre-treated with 3-MA or BECN1 knockdown, reduced the percentage of CHIKV-infected cells, the expression levels of the E1 and C proteins, and the release of viral RNA copies in culture supernatants [161]. Pre-treatment of HEK293 with rapamycin, on the other hand, enhanced the percentage of infected cells, E1 glycoprotein expression and viral RNA in the culture supernatants [161]. Similarly, autophagy induction with rapamycin in HeLa cells significantly increased CHIKV replication and viral production at 15 and 24 hpi as assessed by FACS and $TCID_{50}$, while inhibition using wortmannin, or BECN1 or ATG7 knockdown restricted viral replication [163]. These results indicate that autophagy

**Table 3.** Summary of the literature describing an antiviral or proviral role of autophagy or ATG proteins over the course of CHIKV infection.

| (a) CHIKV | | | |
|---|---|---|---|
| (i) (in vitro) | | | |
| cell type | experimental approach | role of autophagy[a] | references |
| HEK.293 | rapamycin treatment | proviral | [161] |
| | 3-MA treatment | | |
| | BECN1 siRNA | | |
| Atg5−/− MEFs | — | antiviral | [162] |
| HeLa | TAT-BECN1 peptide | antiviral | [135] |
| HeLa | rapamycin treatment | proviral | [163] |
| | wortmannin treatment | | |
| | BECN1, ATG7 and NDP52 siRNA | | |
| | p62 siRNA | antiviral | |
| HLFs | NDP52 siRNA | proviral (NDP52) | |
| | p62 siRNA | antiviral (p62) | |
| WT MEFs | NDP52 siRNA | no effect | |
| Atg5−/− MEFs | — | antiviral | |
| (ii) (in vivo) | | | |
| animal model | experimental approach | role of autophagy[a] | references |
| Atg16l1[HM] mice | — | no effect on viral titres but increased pathogenesis (lethality) | [162] |

[a]Measured by assessing viral titres, percentage of infection, extracellular or intracellular RNA.

induction is beneficial for CHIKV replication and release in human cell lines.

Interestingly, depletion of NDP52, an autophagy receptor involved in several aspects of immunity [167], significantly decreased CHIKV replication, protein translation and viral titres in HeLa cells, suggesting that NDP52 positively controls CHIKV replication [163]. In HeLa cells and primary human labial fibroblasts (HLFs) exposed to CHIKV, NDP52 was shown to co-immunoprecipitate with the nsP2 viral protein and partially co-localized with dsRNA and nsP2, but not LC3, in the perinuclear region [163]. These results indicate that autophagy could promote CHIKV replication in human cells, at least partly, by the interaction of NDP52 with nsP2 near the sites of protein translation. The beneficial role of NDP52 was, however, not reproduced in MEFs, suggesting that CHIKV replicates differently in mouse-derived cells [163].

### 2.5.2.2. Evidence pointing towards an antiviral role of autophagy in CHIKV infection

An antiviral role of autophagy in CHIKV infection is based on the observation of an increased percentage of infected cells and progeny virus titres in $Atg5^{-/-}$ MEFs exposed to MOI 0.1 CHIKV-GFP at 48 hpi [162]. Furthermore, knockdown of p62 in HeLa cells was found to increase viral replication at 15 hpi [163]. Additionally, it was shown that the viral C protein is ubiquitinated and co-localizes with both p62 and LAMP1, suggesting that autophagy, through p62, targets CHIKV components to autolysosomes for degradation to counteract infection [163]. Indeed, the interaction between p62 and the viral C protein was shown by co-immunoprecipitation, and treatment of CHIKV-infected

cells with bafilomycin A1 led to an increase of C protein levels [163]. These results indicate an autophagy-mediated antiviral response against CHIKV, enabled by the recognition of the viral C protein by the p62 receptor, in both human and mouse-derived cells.

### 2.5.3. Autophagy crosstalk with other pathways

CHIKV induces the PERK and the IRE1α arms of the UPR in U-87 MG cells [164]. Furthermore, activation of IRE1α correlated with increased steady-state levels of LC3 lipidation, and siRNA knockdown of IRE1α decreased CHIKV-induced LC3 puncta formation in MEFs [162]. ROS-mediated activation of AMPK has also been observed in CHIKV-infected MEFs, and this coincided with a decrease in mTOR phosphorylation at 24 hpi, which in turn correlated with an increase in LC3-I conversion into LC3-II [162]. Therefore, autophagy initiation is a downstream response to ER and oxidative stress activated by CHIKV replication, possibly through the transcriptional activation of several ATG genes [168,169].

CHIKV induces apoptosis via intrinsic and extrinsic mechanisms [170], which have been suggested to be modulated by autophagy as a mechanism that controls viral pathogenesis early in infection. In CHIKV-infected HFFs and MEFs, it was shown using imaging flow cytometry that autophagy and apoptosis are mutually exclusive processes, as autophagy initiation from 24 to 48 hpi prevented CHIKV-induced caspase 3 activation [162]. Experiments performed by the same authors revealed increased CHIKV-induced lethality in mice carrying an Atg16l1 hypomorphic allele in comparison to the WT mice [162]. This study, however, did not find any differences in the viral titres of

these mice, and they suggested that autophagy does not significantly affect viral infection *in vivo*.

# 3. Conclusion and perspectives: is there an integrated view?

Despite the public health impact of arboviruses transmitted to humans by mosquitoes, treatments and prophylactic measures to combat these viruses remain scarce and therefore the research on the virus–host interactions has intensified in recent years. Overall, DENV, WNV, ZIKV and CHIKV appear to activate autophagy either directly or indirectly through diverse mechanisms. DENV and ZIKV were described to actively initiate autophagy in several cellular models, possibly through the expression of NS4A, NS4B and/or NS1. For WNV, however, autophagy induction is determined by the cell type and the virus strain used. Site-directed mutagenesis studies can be performed to further address the role and importance of the NS proteins in autophagy initiation during flavivirus infection. CHIKV also triggers autophagy in most of the cells types evaluated so far, although it remains to be identified which CHIKV protein is associated with this cellular response. There is also evidence that autophagy is induced through the activation of ER or oxidative stress as a consequence of DENV and CHIKV infection. UPR induction, however, does not always trigger autophagy during arboviral infection as conflicting results have been described for WNV. Furthermore, the initiation of autophagy attenuates cellular stress and was described to prevent cells from undergoing apoptosis. Therefore, at least for DENV and CHIKV, the initiation of autophagy is ultimately beneficial for cell survival and viral replication.

As we have highlighted throughout this review, the measurement of the autophagic flux during arboviral infection is technically challenging. In this regard, early investigations mainly assessed changes in the steady-state levels of LC3-II and p62 and the number of LC3-positive puncta. More recently, the use of bafilomycin A1 and other lysosomal inhibitors, assays based on the GFP–RFP–LC3 tandem construct, and image-based flow cytometry have allowed us to properly measure the autophagic flux. Based on these findings, early in DENV infection autophagosomes are formed, whereas at later time points in infection lysosomal degradation of autophagosomes is impaired. This is in contrast to ZIKV, WNV and CHIKV, for which autophagy induction was shown to culminate in autophagosome degradation.

Once autophagy is triggered, a major research question has been whether it positively or negatively influences arboviral replication. This has represented an additional challenge for the field, as several discrepancies were described, mainly associated with cell type variations or to the methods employed to investigate this subject. In this regard, and as we have also emphasized throughout the review, the use of compounds inducing and inhibiting autophagy-like rapamycin or 3-MA is not optimal, as these molecules are known to affect multiple cellular pathways. For this reason, genetic approaches and methods based on the specific depletion of ATG proteins from different autophagy functional clusters should be the preferred experimental strategy in future research. Most evidence points out that DENV, ZIKV and CHIKV replication is promoted by autophagy, albeit in a cell-type-specific manner. For example, lipophagy has been proven to be beneficial for DENV and ZIKV replication in human hepatic cell lines. In addition, DENV appears to benefit from autophagy during progeny virus particle maturation and spread, but the molecular mechanism remains to be unveiled. For CHIKV, the interaction between nsP2 and the NDP52 autophagy receptor in human cell lines ultimately favours viral replication through an as yet unknown mechanism. Although a proviral function for autophagy has also been suggested for a specific WNV strain, more studies are required to validate these results.

Other evidence suggests that autophagy contributes to the removal of viral components during infection to relieve cellular stress. This phenomenon was described to occur during CHIKV infection in human and murine cells, in which capsid ubiquitination and its co-localization with p62 suggest xenophagic degradation of this viral protein. On the same line, DENV and ZIKV counteract autophagy, underscoring the notion that autophagy acts as an antiviral response. For example, DENV was described to block the autophagic flux by inducing proteosomal turnover of p62. Moreover, DENV, WNV and ZIKV cleave the FAM134B reticulophagy receptor mediated by the NS2B3 protease. Also, studies performed in DENV-infected THP-1 and U937 cells, which both have a monocytic origin, concluded that autophagy has an antiviral function, but the mechanism underlying this role remains unidentified. Different laboratories have attempted to exploit this potential antiviral role of autophagy as a therapeutic strategy. In our view, the most interesting approach constitutes the use of the autophagy-inducing peptide TAT-BECN1 to counteract viral infection and the pathological effects caused by WNV, ZIKV and CHIKV.

In conclusion, autophagy can have diverse outcomes in DENV, WNV, ZIKV and CHIKV infection, and many questions arise from this apparent dual role of autophagy. For example, it remains to be explored how DENV triggers and benefits from lipophagy when the autophagic flux is impaired. In addition, how does DENV prevent the formation of autolysosomal vesicles later in infection? Addressing these and other questions will provide crucial information on the arbovirus replication cycle and the process of autophagy. While studying autophagy, it is also important to consider that ATG proteins participate in many unconventional functions outside the autophagy context [103–105] and different forms of non-canonical autophagy also exist, which has been suggested to contribute to DENV and ZIKV infections [131,171]. One should always compare the results of different methods to firmly conclude whether selective and/or non-selective types of autophagy have a pro- or antiviral role during arboviral infection. Therefore, interdisciplinary approaches involving experts in the field of arbovirus and autophagy may be crucial to design studies for a more thorough understanding of autophagy and arboviruses.

Data accessibility. This article has no additional data.

Competing interests. We declare we have no competing interests.

Funding. F.R. is supported by ZonMW VICI (016.130.606), ZonMW TOP (91217002), ALW Open Programme (ALWOP.310) and Marie Skłodowska-Curie Cofund (713660) and Marie Skłodowska Curie ETN (765912) grants. J.M.S. is also supported by a Skłodowska-Curie Cofund (713660). L.E.-C. is supported by Erasmus Mundus EURICA mobility programme and Graduate School of Medical Sciences PhD scholarship (University Medical Center Groningen).

Acknowledgements. The authors thank Yingying Cong and Nilima Dinesh Kumar for the critical reading of the manuscript.

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
