## [Reviewer comments · Open Biology]

Review History

RSOB-19-0009.R0 (Original submission)

Review form: Reviewer 1

Recommendation

Accept with minor revision (please list in comments)

Are each of the following suitable for general readers?

- a) **Title**
Yes
- b) **Summary**
Yes
- c) **Introduction**
Yes

Is the length of the paper justified?

Yes

Should the paper be seen by a specialist statistical reviewer?

No

Is it clear how to make all supporting data available?

Not Applicable

Is the supplementary material necessary; and if so is it adequate and clear?

Not Applicable

Do you have any ethical concerns with this paper?

No

Comments to the Author

Autophagy is a cellular pathway of fundamental importance for cellular homeostasis as well as nutrient recycling. In recent years it has shown to be important or critical for many normal as well as pathogenic cellular processes. Due to variations in autophagic rates in different cell lines and difficulties in interpretation of multifarious autophagy assays, many conclusions about autophagy in the literature are apparently contradictory.

In this article, the authors review the literature on the regulation of autophagy in selected flavivirus and alphavirus infections. They present a comprehensive overview of the events involved in the initiation and execution of autophagic degradation under normal conditions, before meticulously reviewing the literature on flaviviruses, mainly dengue but also West Nile and Zika virus as well as the alphavirus chikungunya virus. The article is very well researched, well written and will be of great interest to many researchers in the field. Of particular value is the authors attention to the apparently contradictory conclusions in the literature regarding the positive or negative effects of autophagy on viral replication. I have only minor concerns which the authors may wish to address before publication.

Minor issues

Is there some reason why autophagy would be especially relevant for arboviruses? That might explain why the review focuses on the mosquito borne members of the flavivirus and alphavirus genera? By focusing only on these viruses, the authors are excluding much important work on the regulation of autophagy in other related viruses, particularly hepatitis C virus infection. Is there any reason to suspect that the effects of the viruses on autophagy (or vice versa) would be different in mosquito cell lines or indeed in mosquitoes?

There are three main types of autophagy, macro-, micro- and chaperone mediated. This review refers only to macroautophagy as 'autophagy' and ignores the other two types. Is there a reason for this? Have the other two shown not to be important for these viral infections, or has nobody looked?

The authors should be more consistent with capitalisation in some virus names. For example, the first paragraph mentions 'West Nile virus' and 'Chikungunya Virus'. Also, Zika and West Nile should be capitalised because they are proper nouns, but dengue and chikungunya should not.

Page 32, 6 lines from bottom, sentence starting 'This phenomenon...': Care should be taken interpreting results from the Orvedahl paper since they were generated using Sindbis virus

expressing an mCherry-capsid fusion protein. Such a large tag cloned into a protein which multimerises into a tightly packed symmetrical structure, might well be expected, at least to some extent, to misfold, aggregate and be targeted for degradation. Some of their conclusions were challenged by Zheng and Kielian 2013 (PMID: 23785213), using a smaller tag. Also, if results with Sindbis are to be reviewed in CHIKV-relevant context, then the authors might also wish to discuss Eng et al., 2012 (PMID: 22438538), studying the regulation of autophagy in the even more closely related Semliki Forest virus.

Figure 1: If the dimensions of the figure in my PDF are the same to be used in the final version, then the font size is far too small. Also the authors may wish to rethink the use of white text on light blue background which is close to invisible when printed b+w.

Decision letter (RSOB-19-0009.R0)

01-Feb-2019

Dear Professor Reggiori,

We are pleased to inform you that your manuscript RSOB-19-0009 entitled "Role of autophagy during the replication and pathogenesis of common mosquito-borne flavi- and alphaviruses" has been accepted by the Editor for publication in Open Biology. The reviewer has recommended publication, but also suggest some minor revisions to your manuscript. Therefore, we invite you to respond to the reviewer's comments and revise your manuscript.

Please submit the revised version of your manuscript within 14 days. If you do not think you will be able to meet this date please let us know immediately and we can extend this deadline for you.

When submitting your revised manuscript, you will be able to respond to the comments made by the referee(s) and upload a file "Response to Referees" in "Section 6 - File Upload". You can use this to document any changes you make to the original manuscript. In order to expedite the processing of the revised manuscript, please be as specific as possible in your response to the referee.

- 1) A text file of the manuscript (doc, txt, rtf or tex), including the references, tables (including captions) and figure captions. Please remove any tracked changes from the text before submission. PDF files are not an accepted format for the "Main Document".

2) A separate electronic file of each figure (tiff, EPS or print-quality PDF preferred). The format should be produced directly from original creation package, or original software format. Please note that PowerPoint files are not accepted.

3) Electronic supplementary material: this should be contained in a separate file from the main text and meet our ESM criteria (see <http://royalsocietypublishing.org/instructions-authors#question5>). All supplementary materials accompanying an accepted article will be treated as in their final form. They will be published alongside the paper on the journal website and posted on the online figshare repository. Files on figshare will be made available approximately one week before the accompanying article so that the supplementary material can be attributed a unique DOI.

Online supplementary material will also carry the title and description provided during submission, so please ensure these are accurate and informative. Note that the Royal Society will not edit or typeset supplementary material and it will be hosted as provided. Please ensure that the supplementary material includes the paper details (authors, title, journal name, article DOI). Your article DOI will be 10.1098/rsob.2016[last 4 digits of e.g. 10.1098/rsob.20160049].

4) A media summary: a short non-technical summary (up to 100 words) of the key findings/importance of your manuscript. Please try to write in simple English, avoid jargon, explain the importance of the topic, outline the main implications and describe why this topic is newsworthy.

Images

Data-Sharing

It is a condition of publication that data supporting your paper are made available. Data should be made available either in the electronic supplementary material or through an appropriate repository. Details of how to access data should be included in your paper. Please see <http://royalsocietypublishing.org/site/authors/policy.xhtml#question6> for more details.

Sincerely,

The Open Biology Team
<mailto:openbiology@royalsociety.org>

Reviewer's Comments to Author:

Autophagy is a cellular pathway of fundamental importance for cellular homeostasis as well as nutrient recycling. In recent years it has shown to be important or critical for many normal as well as pathogenic cellular processes. Due to variations in autophagic rates in different cell lines and difficulties in interpretation of multifarious autophagy assays, many conclusions about autophagy in the literature are apparently contradictory.

In this article, the authors review the literature on the regulation of autophagy in selected flavivirus and alphavirus infections. They present a comprehensive overview of the events involved in the initiation and execution of autophagic degradation under normal conditions, before meticulously

reviewing the literature on flaviviruses, mainly dengue but also West Nile and Zika virus as well as the alphavirus chikungunya virus. The article is very well researched, well written and will be of great interest to many researchers in the field. Of particular value is the authors attention to the apparently contradictory conclusions in the literature regarding the positive or negative effects of autophagy on viral replication. I have only minor concerns which the authors may wish to address before publication.

Minor issues

Is there some reason why autophagy would be especially relevant for arboviruses? That might explain why the review focuses on the mosquito borne members of the flavi and alphavirus genera? By focusing only on these viruses, the authors are excluding much important work on the regulation of autophagy in other related viruses, particularly hepatitis C virus infection. Is there any reason to suspect that the effects of the viruses on autophagy (or vice versa) would be different in mosquito cell lines or indeed in mosquitos?

There are three main types of autophagy, macro-, micro- and chaperone mediated. This review refers only to macroautophagy as 'autophagy' and ignores the other two types. Is there a reason for this? Have the other two shown not to be important for these viral infections, or has nobody looked?

The authors should be more consistent with capitalisation in some virus names. For example, the first paragraph mentions 'West Nile &v&/u&irus' and 'Chikungunya &V&/u&irus'. Also, Zika and West Nile should be capitalised because they are proper nouns, but dengue and chikungunya should not.

Page 32, 6 lines from bottom, sentence starting 'This phenomenon...': Care should be taken interpreting results from the Orvedahl paper since they were generated using Sindbis virus expressing an mCherry-capsid fusion protein. Such a large tag cloned into a protein which multimerises into a tightly packed symmetrical structure, might well be expected, at least to some extent, to misfold, aggregate and be targeted for degradation. Some of their conclusions were challenged by Zheng and Kielian 2013 (PMID: 23785213), using a smaller tag. Also, if results with Sindbis are to be reviewed in CHIKV-relevant context, then the authors might also wish to discuss Eng et al., 2012 (PMID: 22438538), studying the regulation of autophagy in the even more closely related Semliki Forest virus.

Figure 1: If the dimensions of the figure in my PDF are the same to be used in the final version, then the font size is far too small. Also the authors may wish to rethink the use of white text on light blue background which is close to invisible when printed b+w.

Decision letter (RSOB-19-0009.R1)

18-Feb-2019

Dear Professor Reggiori,

We are pleased to inform you that your manuscript entitled "Role of autophagy during the replication and pathogenesis of common mosquito-borne flavi- and alphaviruses" has been accepted by the Editor for publication in Open Biology.

You can expect to receive a proof of your article from our Production office in due course, please

check your spam filter if you do not receive it within the next 10 working days. Please let us know if you are likely to be away from e-mail contact during this time.

Sincerely,

The Open Biology Team
mailto: openbiology@royalsociety.org